# Application of Dual-Frequency Self-Injection Locked DFB Laser for Brillouin Optical Time Domain Analysis

**DOI:** 10.3390/s21206859

**Published:** 2021-10-15

**Authors:** Cesar A. Lopez-Mercado, Dmitry A. Korobko, Igor O. Zolotovskii, Andrei A. Fotiadi

**Affiliations:** 1Scientific Research and Advanced Studies Center of Ensenada (CICESE), 22860 Ensenada, BC, Mexico; cmercado@cicese.edu.mx; 2S.P. Kapitsa Research Institute of Technology, Ulyanovsk State University, 42 Leo Tolstoy Street, 432970 Ulyanovsk, Russia; korobkotam@rambler.ru (D.A.K.); rafzol.14@mail.ru (I.O.Z.); 3Electromagnetism and Telecommunication Department, University of Mons, B-7000 Mons, Belgium; 4Ioffe Physical-Technical Institute of the RAS, 26 Polytekhnicheskaya Street, 194021 St. Petersburg, Russia

**Keywords:** distributed fiber sensing, BOTDA, Brillouin fiber laser

## Abstract

Self-injection locking to an external fiber cavity is an efficient technique enabling drastic linewidth narrowing of semiconductor lasers. Recently, we constructed a simple dual-frequency laser source that employs self-injection locking of a DFB laser in the external ring fiber cavity and Brillouin lasing in the same cavity. The laser performance characteristics are on the level of the laser modules commonly used with BOTDA. The use of a laser source operating two frequencies strongly locked through the Brillouin resonance simplifies the BOTDA system, avoiding the use of a broadband electrooptical modulator (EOM) and high-frequency electronics. Here, in a direct comparison with the commercial BOTDA, we explore the capacity of our low-cost solution for BOTDA sensing, demonstrating distributed measurements of the Brillouin frequency shift in a 10 km sensing fiber with a 1.5 m spatial resolution.

## 1. Introduction

Distributed optical fiber sensors show superior advantages over their electronic counterparts due to their high sensitivity to external disturbances and low loss transmission, which are important for remote sensing [1]. Unlike single-point optical fiber sensors [2,3], distributed optical fiber sensors can interrogate and spatially resolve measurands along an optical fiber due to their specific sensing mechanisms [4,5,6,7]. Among them, Brillouin-based distributed sensors [8,9,10] have attracted immense interest in recent years in fields such as the health monitoring of large structures in oil and gas pipelines [11], railways and high-voltage transmission lines [12], high-temperature distributed measurement in industrial applications [13], distributed strain measurement for cracks detection [14], and structural health monitoring [15]. The conventional distributed Brillouin optical fiber sensing is based on the backward-stimulated Brillouin scattering, where the strain or temperature is a linear function of the Brillouin frequency shift (BFS) and so can be recovered from the distribution of Brillouin gain spectra (BGS) along the sensing fiber [16,17]. Over the past two decades, many efforts have been devoted to improving its performance, including spatial resolution, measurement time, and sensing range [18,19,20]. Although the Brillouin sensing instruments have become commercially available, their relatively high cost remains the major critical factor limiting the range of their potential applications. A key and most expensive part of the traditional BOTDA system is a master-oscillator module employed for generation and tuning of the pump and Stokes signal frequencies. Commonly, such modules implement phase-locked loop (PLL) [21] or optical side-band (OSB) generation techniques [22]. In the PLL technique, two narrowband laser sources are used to generate pump and Stokes laser frequencies. The frequency of one laser is locked to the frequency of another laser through a feedback circuit, which allows tuning the frequency difference. This technique requires the use of high-frequency photodetectors and RF generators. In the OSB method, only one narrowband laser is used, whereas the second frequency is generated through a broadband electro-optical modulator (EOM) driven at the frequency corresponding to the desired frequency shift (~11 GHz). This technique employs a broadband EOM and high-frequency drivers. Therefore, besides the narrowband laser sources with strict stability requirements, both methods require the use of rather costly high-frequency devices and electronic circuits. Considerable efforts have been directed to simplifying the generation and tuning of pump and Stokes signals. A number of new BOTDA solutions have been proposed. Among them are sensors employing a single optical source driven by pulsed RF signals and passive optical filtering [23], systems based on time-division pump-probe generation by direct modulation of a laser diode through an arbitrary waveform generator [24], and sources that use the Stokes wave generated inside a highly nonlinear fiber [25]. The idea to use a Brillouin fiber laser has been raised and investigated for this purpose [26,27,28]. However, stable lasers are supplied by rather expensive active stabilization circuits. Concerning the capacity of BOTDA systems based on Brillouin laser sources, the previous works were not conclusive and no comparison with the commercial BOTDA systems was performed.

An alternative solution for a dual-frequency laser potentially suitable for BOTDA applications was reported recently [29]. The principle of the laser operation is based on the self-injection locking mechanism, which enables the reduction in the DFB laser linewidth down to the sub-kilohertz level [30]. The ability of this low-cost source to replace the standard laser in the phase-OTDR analyzer has been experimentally demonstrated [31,32]. In the dual-frequency laser configuration [33], the same ring fiber cavity is used to generate narrow-band coherent light at the pump frequency (through the self-injection locking mechanism) and narrow-band coherent light at Stokes frequency (through stimulated Brillouin scattering). The laser is supplied with a simple low-bandwidth active optoelectronic feedback circuit governed by a low-cost USB-DAQ card [34]. Importantly, the drastic narrowing of the DFB laser linewidth to the sub-kilohertz range is provided by the self-injection-locking mechanism, whereas the active feedback is used only to maintain the laser operation in this regime. Therefore, in terms of feedback circuit bandwidth, complexity, and allocated memory, this method is much less consuming than the optoelectronic systems commonly used with Brillouin fiber lasers, including the lasers earlier considered for BOTDA sensing [27].

In this paper, we explore the potential of this low-cost laser solution for operation with the Brillouin optical time domain analyzer (BOTDA) [17]. We present an extended description of the experimental results, evaluating the capacity of the reported laser configuration for distributed measurements of the Brillouin frequency shift (BFS) in a 10 km fiber testing line with a spatial resolution of 1.5 m. The control measurements in the same testing line were performed with a commercial BOTDA set (OZ-optics, OZ Optics Limited, Canada) for investigation using the same spatial resolution. A direct comparison of the results highlights no deterioration in the BOTDA system characteristics associated with the use of low-cost dual-frequency laser. Importantly, the laser source operating at two frequencies strongly locked through the Brillouin resonance simplifies the BOTDA system and excludes the use of a broadband EOM and the high-frequency electronics commonly employed with the BOTDA system based on a single master-oscillator.

## 2. Experimental Setup

Our experimental BOTDA configuration is shown in Figure 1. A simple self-injection locked Brillouin laser operating at 1535 nm [33] was used as a master-oscillator. The laser generated two monochromatic optical waves at pump ($\nu_{L0}$) and Stokes ($\nu_{S0}$) frequencies delivered through two independent fiber outputs. The laser operation is characterized by natural Lorenz linewidths of ~270 and ~110 Hz, and powers of ~9 mW and ~100 μW for pump and Stokes outputs, respectively. The laser optical spectra shown in Figure 2 demonstrate the contrast of ~70 dB between the peak and background powers for both laser outputs; the spectra are centered at 1531.13 and 1531.21 nm, respectively. The spectrum widths are determined by the resolution (~0.01 nm) of the spectrum analyzer. The difference of ~0.08 nm between the peak wavelengths corresponds to a Brillouin frequency shift (BFS) of ~11 GHz. A weak Rayleigh scattering signal at the pump frequency was also observed through the Stokes laser output.

The beating between two laser outputs is characterized by a stable radio-frequency (RF) spectrum with 290 Hz -linewidth and fixed peak frequency at $\nu_{L0}-\nu_{S0}=\Delta \nu_{0}\sim 10.946\;GHz$, as presented in Figure 3a. The peak RF spectrum corresponds to the Brillouin frequency shift in SMF-28 fiber (Corning, Inc., Corning, NY, USA) used in the laser cavity at 1535 nm. To reduce the effect of the environmental noise, a spliced laser configuration was placed into a thermostabilized (∼25 °C) foam box. Additional thermal control was applied to the laser box as a whole and used to keep the feedback circuit within its dynamic range ensuring long-term laser operation stability. The drift in the RF beat recorded each minute for 1 h is shown in Figure 3b. Variations in the RF spectrum peak frequency within a limited range ($\delta \nu_{RF}$ < 5 kHz) reflects the effect of the thermal control. No mode hopping event was observed when the RF beat frequency was within this range. Details on the laser operation and its performance characteristics are provided in Appendix A.

The BOTDA built in the laboratory is a simple modification of the traditional pump-probe setup [35]. The laser radiation at the pump frequency $\nu_{L0}$ passes through the electro-optical intensity modulator (EOM 1) and erbium-doped fiber amplifier (EDFA) to form a periodic train of rectangular pulses with the peak power of ∼300 mW, pulse duration of ∼15 ns, and repetition rate of ~10 kHz. The laser radiation emitted at the Stokes frequency $\nu_{S0}$ passes through the dual-drive electro-optic modulator (EOM 2, SSB-CS EOM, Sumitomo), variable optical attenuator (VOA), and polarization mode scrambler (PMS) to form a ~10 µW CW probe signal at the frequency $\nu_{S}=\nu_{S0}-\delta \nu_{S}$. The frequency tuning of the probe signal is provided by a tunable radio-frequency generator supplying EOM 2 at $\delta \nu_{S}<1\;GHz$. The pump pulses at $\nu_{L0}$ and the CW probe signal at $\nu_{S}$ were introduced into the fiber under test from the opposite fiber ends. Their interaction through the Brillouin process in the fiber under test caused an energy transfer from the pump pulse to the CW Stokes signal, leading to its intensity modulation recorded at the fiber output by the fast photodetector and PC acquisition card. The modulation amplitude (typically <100 nW) of the probe signal is proportional to the local Brillouin gain at the fiber point where Brillouin resonance is achieved. The recorded probe signal traces were averaged over 4096 pump pulses and used to map the distribution of the Brillouin gain over the fiber length. The traces recorded at different $\delta \nu_{S}$ were used to build the Brillouin gain spectrum at each fiber point. Then, these data were mathematically processed to find the position of the Brillouin spectrum peak, i.e., the BFS, in each fiber point. The pump pulse duration of ~15 ns sets the system spatial resolution of ~1.5 m. Notably, the use of the laser source operating two frequencies strongly locked through the Brillouin resonance as a master-oscillator in our BOTDA setup allowed replacing the broadband EOM used in [35] for the frequency shift by ~11 GHz and all high-frequency electronics by more cost-efficient and straightforward counterparts operating in the sub-gigahertz radio-frequency range.

To evaluate the performance of the laser operation with the built BOTDA, we built a fiber testing line similar to that commonly used with BOTDAs for their calibration. The 10 km length testing line shown in Figure 4 comprises two lengths of SMF-28 Corning fiber (9.1 and 1 km) and a length of OFS (G.657) fiber (~50 m) placed between them. The latter includes eleven altered fiber coils of different lengths (0.75–6.0 m) kept at different temperatures. Odd coils were placed into a heat chamber thermostabilized at 60 °C, while even coils were rested at room temperature (~25 °C). The measured distribution of the BFS over the OFS fiber followed the temperature fiber profile with a factor of ~1 MHz/°C.

## 3. Results

The testing experiments were performed with our BOTDA setup and then repeated with the commercial BOTDA system (OZ-optics) set to operate with a pump pulse of 15 ns. Note that the operating wavelength of the commercial BOTDA is 1550 nm, so the BFS data measured with this device are presented here with a scaling factor of 1550/1535. The experimental results are shown in Figure 5 and Figure 6. First, we adjusted the frequency difference between the pump and probe signal $\Delta \nu_{0}+\delta \nu_{S}$ to maximize the signal recorded from the first length of the SMS-28 fiber. In this case, $\delta \nu_{S}\approx 0$, since the frequency difference of the two signals emitted by the laser corresponds to the BFS in the SMS-28 fiber at room temperature (~10.946 GHz). Figure 5a compares the traces recorded with our setup and the commercial BOTDA: both traces exhibit similar behavior, including the gain modulation pronounced in the fiber points with an alternating temperature. The drop in both traces after 9.2 km is explained by pump power loss after splicing. Figure 5b provides more details on the Brillouin gain distribution along the OFS fiber. Both systems well-recognized the fiber segments where the fiber temperature was higher than the room temperature. Since the BFS increases with temperature, the BFS in hot OFS fiber segments is closer to the frequency difference set by the system, leading to an increase in the local Brillouin gain measured at these points.

In order to accurately estimate the BFS distribution along the sensing fiber, the Brillouin gain spectrum (BGS) was measured at each fiber location. At each fiber point, the radio frequency $\delta \nu_{S}$ applied to the dual-drive EOM 2 was swept within the range of 100 MHz with a step of ~2 MHz. Then, the BFS was reconstructed by fitting the recorded data with the Lorentzian profile [17]. Figure 6a compares the BGSs at two OFS fiber points located at 9151 and 9057 m measured with our setup and the commercial BOTDA. Our system well-reproduced the BGS shape recorded with the commercial BOTDA. Notably, the BGS corresponding to the fiber points exposed to the temperature ~25 and ~60 °C are distant by 35 MHz. Figure 6b shows the distributions of the BFS over the OFS fiber: the quality of BSF restoration is rather high. For the fiber segments longer than 1.5 m, both devices produced almost the same distribution, providing a good agreement in absolute peak frequencies varying from 10,880 GHz (at ~25 °C) to 10,915 GHz (at ~60 °C), resolved slopes, and segment positions. All specific fiber segments were well-recognized. For the fiber segment shorter than 1.5 m, our experimental device demonstrated even higher accuracy than the commercial BOTDA operating with the same pulse duration.

## 4. Conclusions

In conclusion, we used a simple dual-frequency laser configuration operated with BOTDA. The use of the laser operating two fixed frequencies significantly simplified the BOTDA system and allows excluding the broadband EOM (and high-power high-frequency microwave generator supplying the EOM) from the traditional BOTDA configuration based on a single narrowband laser source. These relatively high-cost devices constitute a significant part of the BOTDA system value. Additionally, the laser operation mechanism enabling self-stabilized generation of the Brillouin wave does not employ rather an expensive wavelength locking circuit, being a part of the Brillouin lasers offered for BOTDA sensing earlier [27]. Instead, a dual-drive EOM driven by a standard 1 GHz radio-frequency generator is employed in our home BOTDA setup. Moreover, the tuning range of 100 MHz was enough to perform the measurements shown in Figure 6. Under control of the commercial BOTDA, we evaluated the capacity of a low-cost laser solution to operate with the BOTDA sensing, demonstrating the distributed measurements of the BFS in a 10 km sensing fiber with a 1.5 m spatial resolution. No deterioration of the system performance characteristics associated with the use of the dual-frequency laser was found during the measurements. Further research will be directed to design and testing of new laser sources [36] for advanced sensor applications.

## Figures and Tables

**Figure 1 sensors-21-06859-f001:**
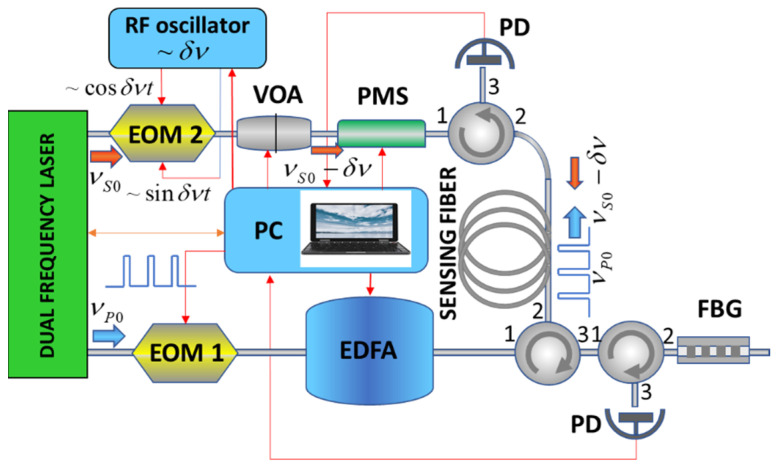
The experimental BOTDA setup.

**Figure 2 sensors-21-06859-f002:**
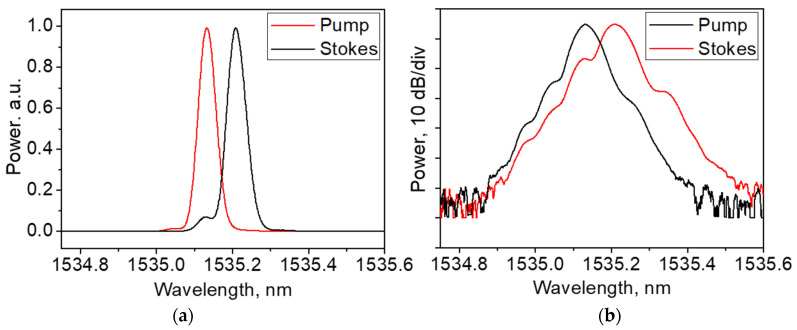
Optical spectra recorded at two laser outputs in (**a**) linear and (**b**) decibel scales.

**Figure 3 sensors-21-06859-f003:**
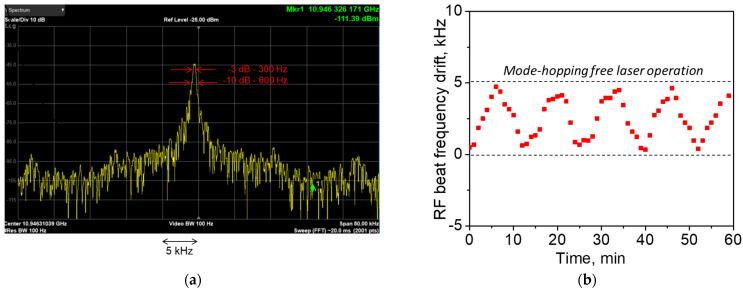
(**a**) Typical RF beat note spectrum measured with two master-oscillator outputs; (**b**) drift in the RF beat frequency measured each minute for 60 min.

**Figure 4 sensors-21-06859-f004:**
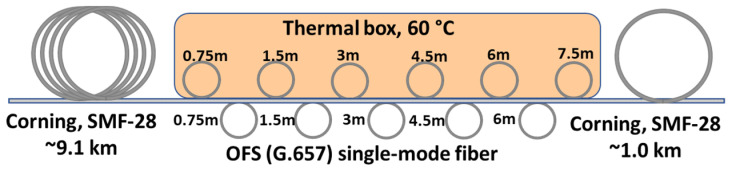
Optical fiber line for BOTDA testing.

**Figure 5 sensors-21-06859-f005:**
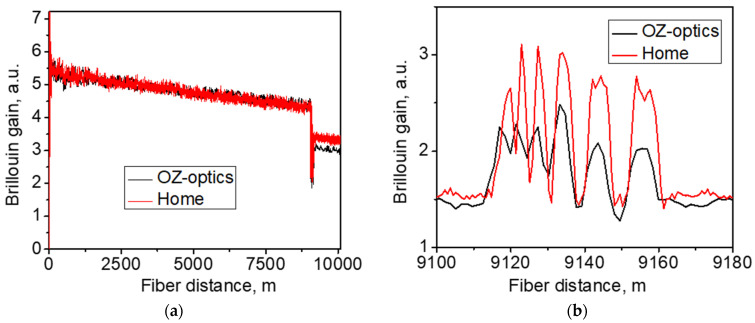
The measured distribution of the Brillouin gain (**a**) over the whole testing line and (**b**) over the range of 9.1–9.18 km.

**Figure 6 sensors-21-06859-f006:**
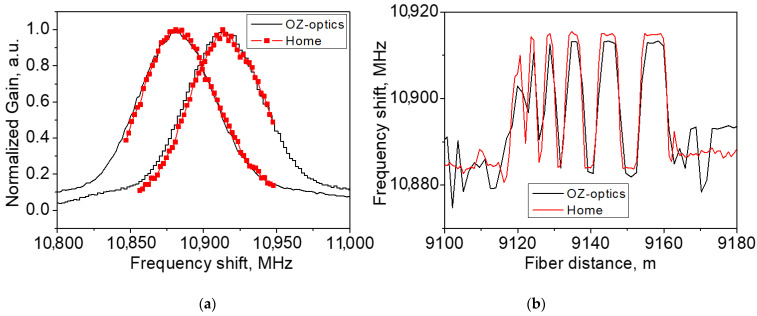
(**a**) The Brillouin gain spectra (BGS) measured at the fiber points of 9151 and 9157 m exposed to the temperature of ~25 and ~60 °C, respectively; (**b**) the measured distribution of the Brillouin frequency shift (BFS) over the range of 9.1 to 9.18 km.

## Data Availability

Not applicable.

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
