# Peer review of "Application of Dual-Frequency Self-Injection Locked DFB Laser for Brillouin Optical Time Domain Analysis"

_sensors, 2021, doi:10.3390/s21206859_

Round 1

Reviewer 1 Report

The paper presents the application of dual-frequency self-injected DFB laser for BOTDA-based distributed optical fiber sensing. As the author mentioned, the laser is well-documented in the previous work: references 10, 13  and a recent conference paper: Proceedings Volume 11772, Optical Sensors 2021; 1177207 (2021) https://doi.org/10.1117/12.2589112. The authors use the same ring fiber cavity (DFB laser diode and passive fiber optics components) to evaluate the capacity of the reported laser configuration for distributed measurements of the Brillouin shift now for BOTDA systems. However, I cannot see a  significant novelty concerning the former work and I wonder why the conference paper is not cited here.

Line 73.The common abbreviation for variable optical attenuator is VOA instead of VAT

Line 92. The caption of Figure 3 is incorrect.

Author Response

Please, see the attached Report Notes

Reviewer 2 Report

This paper presents a low-cost BOTDA system by employing dual-wavelength self-injection locked DFB Laser, and avoid high bandwidth intensity MZM, and RF frequency synthesizer. The paper is interesting of using a dual-wavelength laser, one of the output lies in the Brillouin Stokes region. This can enable a low-cost BOTDA system significantly. However, some of the major comments/concerns are;

  1. Can you show the two optical spectrums of the DFB laser with their SNRs?
  2. The major concern is the stability of the laser. Even a small frequency drifts, then no longer generate stimulated Brillouin scattering. Can you show the figure, which appears the laser peak wavelength over time, at least for one hour?
  3. In the experimental figure, the Circulator arrows are confusing. Can you number the circulator ports with proper lasing directions? How can be the pump pulses side circulator have 4 ports?.
  4. What is the frequency tuning range of the RF generator of the EOM2 to map the three-dimensional gain spectrum?.
  5. The Introduction section needs to be extended with proper literature.
  6. Double check the format of the references.
  7. There are many grammatical errors throughout the manuscript. For instance, “Figure 3, caption”.

Author Response

Please, see the attached Report Notes

Reviewer 3 Report

The manuscript propose dual-frequency felf-injection DFB  Laser for BOTDA-based distributed optical fiber sensing. The structure of manuscript is confuse. The authors needs to rewrite the manuscript and submit a new version. For example: 

  • The references for the first paragraph needs to be changed.
  • The authors in reference 10 aren't the same. Is necessary change the sentence in lines 37-38.

Is necessary a comparasion of the system with others (not only one).

Aplications is necessary.  In title the authors refers to distributed optical fiber sensing but don't show results. 

In present form i recommend reject.

Author Response

Please, see the attached Report Notes

Round 2

Reviewer 1 Report

The contents and the clarity of the  paper are much improved in the revised version.

Author Response

Point 1. The contents and the clarity of the paper are much improved in the revised version.

 Response 1. Thank you very much for your valuable contribution.

Reviewer 2 Report

The authors addressed all the comments and concerns. The manuscript is greatly enhanced, and now it is suitable for publication. 

  1. The experimental setup figure needs to be modified. The electrical connections are confusing, for instance, the detected Brillouin signal looks like fed to the EDFA. Number the EOM, and add radio-frequency generator block to EOM2, etc,.

Author Response

Point 1: The authors addressed all the comments and concerns. The manuscript is greatly enhanced, and now it is suitable for publication.

Response 1. Thank you very much for your valuable contribution.

Point 2: The experimental setup figure needs to be modified. The electrical connections are confusing, for instance, the detected Brillouin signal looks like fed to the EDFA. Number the EOM, and add radio-frequency generator block to EOM2, etc,.

Response 2: Done.

Reviewer 3 Report

The atuhors in reviewed version clarify several points. However, before publish, is possible increase de quality the manuscript.

  • the position of differents figures in manuscript is confuse. The comments to figures appears several paragraph after.
  • some information in Apnedix can be introduced in the main text. 
  • comparation with differents system or similar is necessary.

I recommend a major review.

Author Response

Point 1: The authors in reviewed version clarify several points. However, before publish, is possible increase de quality the manuscript.

Response 1. Thank you very much for your valuable contribution.

Point 2: - the position of differents figures in manuscript is confuse. The comments to figures appears several paragraph after.

Response 2. Revised.

Point 3: - some information in Apnedix can be introduced in the main text.

Response 3. The section “A.2.1. The output powers and optical spectra” (in the part concerning the optical spectra) and section “A.2.6. The RF beat note spectrum and long-term laser stability” are replaced and adapted to the main text.

Point 4: - comparation with differents system or similar is necessary.

Response 4. To meet this concern, the following sentences have been added (revised) to the conclusion:

“The use of the laser operating two fixed frequencies significantly simplifies the BOTDA system and allows to exclude the broadband EOM (and high-power high-frequency microwave generator supplying the EOM) from the traditional BOTDA configuration based on a single narrowband laser source. These relatively high-cost devices constitute a significant part of the BOTDA system value. Besides, the laser operation mechanism enabling self-stabilized generation of the Brillouin wave does not employ rather an expensive wavelength locking circuit being a part of the Brillouin lasers offered for BOTDA sensing earlier [27]. Instead, a dual-drive EOM driven by a standard 1-GHz radio-frequency generator is employed in our home BOTDA setup. Moreover, the tuning range of 100 MHz has been enough to perform the measurements shown in Figure 6. ”

Round 3

Reviewer 3 Report

I agree with the changes. The manuscript can be publish